# In Situ Analysis of DNA-Protein Complex Formation upon Radiation-Induced DNA Damage

**DOI:** 10.3390/ijms20225736

**Published:** 2019-11-15

**Authors:** Giulio Ticli, Ennio Prosperi

**Affiliations:** 1Istituto di Genetica Molecolare “Luca Cavalli Sforza”, Consiglio Nazionale delle Ricerche (CNR), 27100 Pavia, Italy; giulio.ticli@igm.cnr.it; 2Dipartimento di Biologia e Biotecnologie, Università di Pavia, 27100 Pavia, Italy

**Keywords:** DNA damage, DNA repair, DNA-interacting proteins, in situ analysis, immunofluorescence detection, live-cell imaging

## Abstract

The importance of determining at the cellular level the formation of DNA–protein complexes after radiation-induced lesions to DNA is outlined by the evidence that such interactions represent one of the first steps of the cellular response to DNA damage. These complexes are formed through recruitment at the sites of the lesion, of proteins deputed to signal the presence of DNA damage, and of DNA repair factors necessary to remove it. Investigating the formation of such complexes has provided, and will probably continue to, relevant information about molecular mechanisms and spatiotemporal dynamics of the processes that constitute the first barrier of cell defense against genome instability and related diseases. In this review, we will summarize and discuss the use of in situ procedures to detect the formation of DNA-protein complexes after radiation-induced DNA damage. This type of analysis provides important information on the spatial localization and temporal resolution of the formation of such complexes, at the single-cell level, allowing the study of heterogeneous cell populations.

## 1. Introduction

Cells in the human body are continuously exposed to a multitude of endogenous and exogenous agents, which can produce a broad range of DNA lesions, compromising the cell functionality. As a consequence of inefficient or absent DNA repair, DNA damage can lead to genome instability and mutation in DNA [1]. These defects may result in developmental abnormalities and/or in the cell transformation towards a malignant phenotype [2,3].

Radiations (e.g., UV light, X-rays, γ-rays) are one of the major exogenous sources of DNA damage producing different types of lesions, according to the wavelength and the physical nature (electromagnetic or particle) of the radiation. UV radiation is divided into three spectral regions (UV-A: 320–400 nm; UV-B: 290–320 nm; UV-C: 100–290 nm), with UV-B and UV-C being the most dangerous because they are able to induce nucleotide structural alterations, such as the dimerization of pyrimidine bases to form cyclobutane pyrimidine dimers (CPDs) or pyrimidine (6-4) pyrimidone photoproducts (6-4PPs) [4]. UV-A radiation may induce oxidatively generated DNA damage through photosensitization [4]. The more energetic ionizing radiation (IR) induces DNA damage through ionization and hydroxyl radical production, leading to the formation of DNA breaks, either at the single-strand (SSBs) or at the double-strand (DSBs) level, in addition to base oxidation and DNA-protein crosslinks [5,6]. In fact, exposure to IR is commonly used for the therapy of many types of tumor. Given that, the biological effects of radiation have been extensively studied to understand how the cell response to DNA damage may influence the efficacy of radiation therapy [7].

The mechanism by which cells cope with DNA damage has been, and still is, intensively investigated, for the reasons stated above. In the last decades, it has been shown that cells rely on several processes to deal with DNA damage, which now collectively constitute the DNA damage response (DDR) [8]. This term refers to pathways allowing cells to detect the occurrence of DNA lesions, signal their presence to arrest the cell cycle and promote the activation of specific DNA repair systems able to remove such lesions [9]. Preferential activation of DNA damage signaling and repair systems has been shown to directly depend on the type of lesion and the physiological status of the cell [10].

Independently from the DNA damage signaling and repair system considered, the DNA-protein complex formation represents a fundamental step of this intracellular cascade of events because it is required for the DNA lesion recognition and to trigger the recruitment of specific DNA repair factors, which take turns with a defined spatiotemporal sequence during the DNA repair event. Thus, the multiple proteins required for DNA damage signaling and repair need to bind to the DNA for carrying out their activity [11,12]. From early studies using in vitro approaches based on purified proteins in reconstituted systems, components of DDR pathways were identified. However, these methodologies were later considered incomplete and insufficient to address specific questions regarding the spatiotemporal characterization of these processes, taking into account the physiology and ultrastructure complexity of the cell [13]. For this reason, it has been necessary to develop new strategies to probe the formation of DNA-protein complexes in situ, to investigate the recruitment kinetics of DDR proteins at DNA damage sites, and to evaluate where they interact with DNA. These parameters may provide relevant information about the dynamics of these important cellular reactions.

The goal of this review is to cover and discuss the use of the most recent and sensitive methods and techniques to determine the formation of radiation-induced DNA-protein complexes at the cellular level. In fact, the in situ analysis is intended to detect these interactions and related biological processes in a cellular environment that is retained, as much as possible, close to the in vivo condition. A comparison of these techniques in terms of the advantages and limitations of each approach will be discussed.

## 2. In Situ Detection of DNA-Protein Complex Formation

The analysis of DDR factors binding to DNA, and in particular to DNA lesions, may be performed in situ, i.e., in intact cells, either: (i) after appropriate cell/tissue fixation that is necessary to retain these proteins at their activity site, or (ii) directly on living cells, providing that DNA and proteins of interest may be visualized under the microscope (e.g., through protein expression with a fluorescent tag). These methods have been often applied in combination with procedures enabling the induction of localized DNA damage.

The first approach requires an immunolabeling reaction (usually with fluorescently conjugated antibodies) directed to the protein of interest, when it may be identified as bound to DNA. Different procedures are available to perform such analysis, which in general consists of an in situ extraction of the unbound fraction so that only the DNA-bound molecules are subsequently fixed at the sites where DNA lesions are present [14,15,16].

The second approach is based on the more sophisticated technology of live-cell imaging through which the behavior of a DNA-binding protein, e.g., its relocation to DNA damage sites, may be monitored in living cells thanks to its expression in a form fused to a fluorescent tag, such as the green fluorescent protein (GFP) [17]. This technique enables the visualization of the protein of interest at on-going DNA repair sites, thus allowing direct spatiotemporal evaluation of its binding to DNA [18,19,20,21,22].

A schematic overview of both approaches is shown in Figure 1.

### 2.1. Detection of DNA-Protein Complexes in Fixed Cells

#### 2.1.1. Immunofluorescence-Based Techniques

As stated above, a major caveat for this type of analysis is that immunofluorescence reaction must detect the protein of interest when it is bound to DNA (many authors referred to this association as “chromatin-bound fraction”). A practical approach introduced for different proteins known to bind DNA has been to rely on a fixation procedure that may distinguish the DNA-bound form. This has been the case for the “proliferating cell nuclear antigen” (PCNA), an important protein functioning as a platform for many DNA-binding factors participating in several DNA metabolic processes, including DNA replication and repair [23,24]. PCNA encircles the double helix when loaded onto DNA, and it is thus resistant to extraction [14,15]. However, for other proteins, and often for PCNA too, this procedure could not provide reliable information, given that simple fixation may retain part of unbound protein [15,25]. Thus, in situ detection has been most often performed after a detergent-based lysis step enabling the extraction of the unbound fraction while retaining proteins associated with DNA [26].

A significant improvement in the in situ detection of proteins bound to DNA has been obtained by limiting DNA damage to a small area of the nucleus, thereby ensuring that protein binding is detected only at this site. This condition was attained by coupling the immunofluorescence assay to an irradiation procedure that produces localized DNA damage in the nucleus of irradiated cells. In particular, cells were masked with a polycarbonate isopore membrane filter (pore size may be chosen from 3 to 8 µm), allowing the irradiation of defined areas in correspondence of its pores [27,28]. These filters have been extensively used in UV-induced DNA damage studies taking advantage of the physical properties of polycarbonate, which can efficiently absorb the entire spectrum of UV radiation. This technique, named micropore irradiation, has been employed to detect the recruitment of several proteins at local DNA damage sites. Typically, DNA binding factors, such as DDB2 and XPC, that recognize lesions on DNA during nucleotide excision repair (NER), PCNA (Figure 2), as well as repair enzymes, e.g., DNA nucleases, helicases, polymerases, ligases, and chromatin assembly factors, have been detected with this method [29,30,31,32,33,34,35,36].

This protocol of irradiation has allowed the in situ detection of DNA binding of DDR factors, such as p53, at sites of IR-induced DNA damage [37]. The same procedure was used to rule out the presence of this protein at DNA damage sites after UV radiation [38]. In fact, in another study, p53 appeared to localize at sites of UV-induced DNA damage [39]; however, this result was obtained without the extraction of the unbound protein. It must be noted that an important control to safely establish the association of a given protein to DNA in situ, is based on a DNase I digestion step before fixation, which will remove the DNA-protein complex [37].

Other important applications of this technique include the detection of structural components of chromatin that interact with DNA, and how they are involved in the DDR. For instance, the study of histone deposition and chromatin reassembly after DNA damage has taken advantage of the micropore filter irradiation to demonstrate that after UV irradiation, new histone incorporation occurs at damaged sites [40]. In this study, a control DNase treatment was performed to verify that new histone H3 deposition corresponded to the chromatin-bound fraction [40]. Another aspect investigated at local sites of DNA damage is the maintenance of epigenetic information after DNA damage. Acetylation of histone H3 at K56 (H3K56ac) has been recently detected in situ after UV-induced DNA damage [41], thus showing the possibility to study post-repair events at local sites in chromatin.

A possible critical aspect of this technique concerns the optimal localization of DNA repair proteins by immunofluorescence staining, since their visualization may be challenging according to the cell line and the protein itself. Given that, protein-specific extraction protocols were developed to improve the fluorescence signal. In particular, they were based on the combination of different detergent extraction procedures (e.g., Triton X-100 and Igepal) with diverse fixative solutions (e.g., ethanol, methanol-acetone, and formaldehyde), to stain only the chromatin-bound fraction of the protein of interest [25,26,27,32]. For instance, for proteins, such as DDB2 and XPG, a protocol performing detergent lysis simultaneously with fixation, was introduced to enable detection of these factors that were otherwise lost, or their amount reduced [32,42]. More recently, visualization at local DNA damage sites of the above factors, including XPC protein, has been improved by using a mild DNase I digestion before the immunoreaction. This procedure probably acts by making the epitope more accessible to the antibodies and has proved to work with different fixatives [43].

The use of polycarbonate filters is, however, limited to UV radiation because of its insufficient shielding properties (e.g., to X-rays and near-infrared light) [44]. Another issue of the technique is that the position of irradiated areas in the nucleus is randomly distributed, and hence specific nuclear regions (e.g., heterochromatin) cannot be selected before irradiation. Since the timing of DNA repair also depends on local chromatin compaction, the information obtained with this approach is, therefore, limited to the detection of a given protein at the irradiated area (a single frame of the overall process). In later studies, the micropore filter has been coupled with confocal microscopy imaging to irradiate living cells and track in real-time fluorescence-tagged proteins recruited at the DNA damage sites [28].

#### 2.1.2. Probe-Based Techniques

Investigating the interaction among proteins and DNA in situ with conventional immunofluorescence techniques has the typical limitation imposed by the optical resolution of the microscope system. To cope with this problem, the proximity ligation assay (PLA) was developed as a sensitive and reliable method for the in situ detection of two molecules in close proximity to each other (<40 nm) [45].

This approach is based on a pair of oligonucleotide-conjugated probes that can, directly or indirectly, bind two molecules that are supposed to interact. The two oligonucleotides are designed to hybridize to two connector oligonucleotides, which are then ligated to form a circular DNA molecule only when the two probes are spatially close. The circular DNA is successively used as a template for the rolling circle amplification (RCA) reaction, together with fluorescently labeled nucleotides as substrate. Therefore, this system detects as a fluorescent spot, two molecules that are spatially close enough to allow the reaction to occur, thereby leading to the assumption that they are interacting [45]. The sensitivity and reliability of PLA are linked to the nature of the probes: The applications in situ are generally based on the use of two primary antibodies, characterized by binding to the target molecules with high specificity, and two species-specific, oligonucleotide-conjugated secondary antibodies (as probes). This two-step procedure guarantees the reduction in potential PLA binders cross-reactivity, which might be a serious concern especially when the target molecule is not abundant or when the antibody dissociation constant (Kd) for the intended molecule is slightly different from the one for the cross-reactive molecule at similar concentrations [46].

In the last decade, PLA has been improved thanks to the development of new variants of the method that has been optimized according to the experimental needs, for instance, by enhancing probe specificity [47]. PLA has also been coupled with other techniques, such as flow cytometry, to provide subcellular localization assay (SLA), both enabling high-throughput data on spatial information and fluorescence quantitation [48]. A multi-well based platform has also been developed, under the name of “high-throughput imaging PLA (HiPLA), using a library of antibodies to probe nuclear proteins, and their associated post-translational modifications, with the nuclear lamina [49].

In principle, the amplification system of PLA is suitable for any study aiming to investigate protein–protein or DNA–protein complex formation, taking advantage of the large variety of available antibodies raised against specific target molecules and their post-translationally modified forms (e.g., acetylation, ubiquitination, phosphorylation). Recently, this technique was adapted for the use of non-adherent cells [50,51], further increasing the range of possible applications. In fact, PLA has been employed to investigate the binding of known DDR factors, such as MDC1, 53BP1, as well as histone γ-H2AX to DSB [52], as well as new proteins, such as EXOSC10 [53], the androgen receptor variants [54], and karyopherin-α2 [55], providing important insight into their activity during DNA repair. In this regard, new PLA-based techniques have been developed, such as DI-PLA (DNA damage in situ ligation followed by proximity ligation assay) [56] and SIRF (in situ protein interactions at nascent and stalled replication forks) [57]. With DI-PLA, it is possible to localize with high accuracy DSBs at the single-cell level. In the original paper, DNA breaks were induced by exogenous expression of a restriction enzyme; however, the procedure may be easily used in cells exposed to radiation. After DNA damage, T4 DNA polymerase and T4 ligase enzyme are used, respectively, to blunt DSB ends and to ligate with high specificity a biotinylated oligonucleotide. PLA is then carried out using an antibody against biotin and a partner antibody against a DDR marker (e.g., γ-H2AX or 53BP1). In this way, DI-PLA allows the amplification of the fluorescence signal and, therefore, the detection in situ of a single DSB in proximity to a DDR protein, overcoming the necessity to use high levels of DNA damage [58].

A particular use of PLA for DNA–protein interaction has been proposed for the detection of protein binding to specific DNA sequences, by using DNA-binding probes for in situ hybridization, in combination with an antibody to a target protein. Circularization oligonucleotides are then hybridized to both PLA probes, ligated, and RCA reaction performed as in normal PLA [59]. This technique may thus be employed for studying protein binding to specific nuclear sequences after DNA damage.

The variation of the PLA method, named SIRF, was developed to study the protein and DNA–protein complexes at active and stalled replication forks at a single-cell level [57]. This method draws its inspiration from another important technique, the iPOND assay (isolation of proteins on nascent DNA) [60]. In particular, SIRF starts with the incorporation of the nucleoside analog 5′-ethylene-2′-deoxyuridine (EdU), which, once incorporated into DNA, may be biotinylated using click chemistry. After that, PLA is carried out using specific antibodies to biotin and a protein target of interest (e.g., 53BP1) [57], as described above. This technique is unique in that it detects the association of proteins with nascent DNA in situ, providing high temporal resolution, in addition to spatial information. In fact, the EdU labeling time is short enough (8 min) to infer that the protein binding to DNA occurs at the DNA synthesis site. Therefore, in this context, both DI-PLA and SIRF are efficient molecular tools for qualitative and quantitative studies on DNA repair systems providing in situ analysis with high resolution.

### 2.2. Detection of DNA–Protein Complexes in Living Cells

The techniques above described are characterized by limited spatial resolution, or do not allow the kinetic analysis of the protein binding to DNA, because they provide a freeze-frame; therefore, they are unsuitable for studying the dynamic binding of repair factors to DNA in living cells. These limitations were overcome by the advent of fluorescent proteins (e.g., GFP), and their derivatives, that may be expressed in living cells, thus making possible real-time studies [18,19,20]. This methodology was further used in combination with the use of a laser beam to produce DNA lesions in a sub-nuclear area. In fact, a focused laser beam allows micro-irradiation with high spatial resolution, in the range of a few microns (≈2 μm), a nuclear region of interest (ROI), thus producing local DNA damage (Figure 1). This approach was initially developed to study UV-induced DNA damage [61], but it was then adapted to induce SSBs and DSBs by cell pre-treatment with sensitizers, such as DNA dyes (Hoechst 33342) and halogenated thymidine analogs (BrdU, IdU) [19,62]. These compounds are less frequently used for this purpose because of their toxic effect on cells due to the oxidatively generated DNA damage induced during photosensitization, and because they may trigger multiple DNA repair processes [63].

One of the most relevant advantages of this approach is that visualization of DNA–protein complexes is easily accomplished with the use of plasmid vectors driving the transient or constitutive expression of a protein of interest, fused to a fluorescent tag. Therefore, the fluorescence live-cell imaging conditions combined with the laser micro-irradiation systems, represent a powerful tool for the in situ studies. This system allows monitoring in real-time the recruitment and subsequent release of DNA binding protein upon the DNA lesion formation. Crucial insights are obtained into the dynamics of protein–DNA interaction turnover, as well as the simultaneous analysis of multiple factors at DNA damage sites when they are available and expressed as fluorescent proteins with different color emission. Along with this, other methods, such as fluorescence recovery after photobleaching (FRAP) and fluorescence loss in photobleaching (FLIP), have been employed after laser micro-irradiation to further analyze and refine kinetic models of DNA-protein interaction, as for DNA damage signaling and repair factors [64,65].

Live-cell imaging has been extensively used to investigate DNA damage signaling pathways and DNA repair processes, and many examples are reported in relevant and recent reviews [12,19,65,66]. Of particular interest are those studies in which quantitative determination of the recruitment and release kinetics at the DNA damage sites was performed for direct players and regulators of the various DNA repair processes. Specific parameters were estimated during NER, (i.e., XPA, XPC, PCNA, p21) [67,68,69,70], in the base excision repair (BER) process (i.e., XRCC1, OGG1, PARP-1) [71,72,73,74,75,76,77], as well as in the non-homologous end joining (NHEJ) process (i.e., Ku70/80 complex, DNA-PK, DNA Ligase III) [78,79,80]. Furthermore, calculation of several kinetic parameters, such as the ratio between protein-bound state and diffusive state over time, allowed the further deepening of theoretical models regarding DNA–protein, or protein–protein complex formation upon DNA damage, and the evaluation quantitatively of the spatiotemporal activity of DNA repair factors [67,69,71,75].

A particular approach in detecting DNA–protein complexes consists of the expression of a fluorescent reporter (i.e., a GFP-labeled peptide or protein) that can bind DNA specifically to a given lesion (e.g., DSB), so that fluorescent localization of DNA damage is obtained in living cells [81]. Expression of the fluorescent GAM protein, or a peptide from 53BP1 protein, have provided localization of DSBs [82,83]. Very recently, UV-induced DNA damage recognition by fluorescently-tagged enzymes (photolyases), specifically binding to CPDs and 6-4PPs, has been proposed [84]. This technique provides a useful tool to detect these lesions in situ and to quantify their repair kinetics in real-time.

To study DNA–protein interactions in the other way round, the expression of a fluorescently-labeled plasmid containing a single lesion (e.g., an abasic site) has been recently proposed for monitoring the recruitment of endogenous proteins to specific lesions over time [85]. However, while the plasmid localization may be performed in vivo, detection of the protein is obtained after fixation and immunostaining to determine the co-localization of the fluorescent plasmid and the protein of interest.

Another interesting method allows the targeted disruption of protein interactions in living cells [86]. The procedure consists in anchoring to a defined cellular structure (e.g., a specific DNA sequence or regions) a GFP-fusion protein that interacts with a partner labeled with a different fluorescent protein (e.g., RFP), and monitoring the recruitment of the red protein to the site of the GFP-tagged one. This method may be applied under different conditions, including treatment with agents able to disrupt the interaction between the two proteins, thus providing important information on their association and the possibility to test the effect of small molecules, drugs, and even peptides [86].

A peculiarity of live-cell imaging techniques is that DNA lesions may be produced in specific nuclear regions. For instance, they have been used to discriminate the recruitment of DNA repair proteins at damaged sites in heterochromatin vs. euchromatin regions: light-stimulated production of oxidatively generated DNA lesions was obtained with Killer Red protein fused to transcription repressor, or activator cassettes integrated into distinct genomic loci [87]. Lastly, the chromatin unfolding effect induced by PARP-1 on the accessibility of DNA-binding vs. histone binding domains to DNA damage sites has been investigated by live-cell imaging and fluorescence lifetime measurements [88].

Using live-cell imaging techniques, software (e.g., ImageJ) for data processing play a fundamental role in imaging acquisition and analysis. In fact, they are helpful in compensating for common experimental issues, such as cell movement, which, especially during long time-lapse, may affect the accuracy of the analysis. This would bring about the inaccurate estimation of fluorescence intensity variations over time, which is at the basis of every kinetic parameter calculation. In particular, this experimental issue would affect the analysis because of the progressive misalignment along the frame sequence between the irradiated spot within the cell and the ROI selected for quantitative fluorescence analysis. However, cell movement during time-lapse experiments can often be corrected by using appropriate plugins (e.g., TurboReg, StackReg), which can align or match images taking advantage of landmarks manually or automatically selected on the image used as a template.

Recent studies have demonstrated that continuous wave (CW) diode lasers, commonly used in confocal microscopy (emitting at 405, 488, or 561 nm), can induce different types of DNA damage (e.g., CPD, SSB, DSB) according to the laser wavelength and to the amount of energy locally deposited, even in the absence of a pre-treatment with thymidine analogs [63,89,90]. In addition, exposure to high-intensity UV laser (266 nm) may induce base ionization and formation of oxidized bases [91]. Thus, a major caveat when using laser micro-beams is that it is not always possible to activate a specific DNA repair process exclusively, and careful evaluation of experimental conditions is to be taken into account. However, the induction of highly-localized DSBs with heavy ions overcomes this limitation, allowing high-resolution fluorescence imaging in living cells [92].

An important common issue occurring in live-cell imaging is photobleaching, a well-known phenomenon in fluorescence microscopy consisting of the gradual reduction of overall fluorescence intensity within the cell. It occurs because of fluorophore exposure to photon-induced chemical damage produced by light that finally leads to a reduction in the resolving power of the fluorescence microscope [93,94,95]. Another important light-induced effect that should be taken into account is phototoxicity arising from photophysical and photochemical processes leading to cellular damage [96,97]. In fact, the local formation of reactive oxygen species (ROS) can damage or destabilize biomolecules other than DNA. In addition, thermal effects due to local energy deposition can lead to protein denaturation [98]. Given that, live-cell imaging techniques should be accompanied by phototoxicity assessment to estimate the light-induced cellular damage, which may affect data reliability and reproducibility. Several biological criteria and physical parameters for phototoxicity evaluation have been identified, as discussed in detail in [99]. Nowadays, different solutions are available to reduce photobleaching and phototoxicity detrimental effects, such as the removal of vitamins riboflavin and pyridoxal from imaging medium [100,101], or the use of antioxidants to increase the chromophore photostability [102]. However, the most promising results have been obtained by optimizing the microscope illumination system to reach the best compromise between the amount of energy released and the fluorescence signal quality [103,104,105].

## 3. Conclusions

Investigations into radiation-induced DNA damage have led to the development and exploitation of several methods to detect DNA–protein interaction in situ. These studies have been fundamental for the understanding of several aspects of DDR, particularly for unraveling complex networks of protein binding to DNA for damage signaling and repair [8,9,10,11,12].

The different approaches in fixed vs. living cells must take into account that in the former, information obtained is more often only qualitative, providing localization, and at most temporal description, of the association of a given protein to DNA (e.g., at damaged sites). However, the use of the micropore filter irradiation technique is simple and does not require expensive instrumentation. Similarly, PLA and SIRF techniques have gained success thanks to the resolution provided, and for the relatively simple methodology. In contrast, the approach on living cells, such as fluorescence live-cell imaging, is undoubtedly a cutting-edge method providing high spatiotemporal resolution and is an essential tool for in situ kinetic studies. The possibility of measuring the fluorescence signal permits the quantification of the process in real-time, thereby allowing the estimation of important kinetic parameters of the reaction underlying the formation of the DNA–protein complex. In the face of these features, live-cell imaging requires more complex and expensive instrumentation.

In conclusion, the possibility to rely on different techniques, ranging from simple to more sophisticated ones, allows researchers to tackle biological problems from different points of view, to gain more insights into the study of radiation-induced DNA-protein interactions.

## Figures and Tables

**Figure 1 ijms-20-05736-f001:**
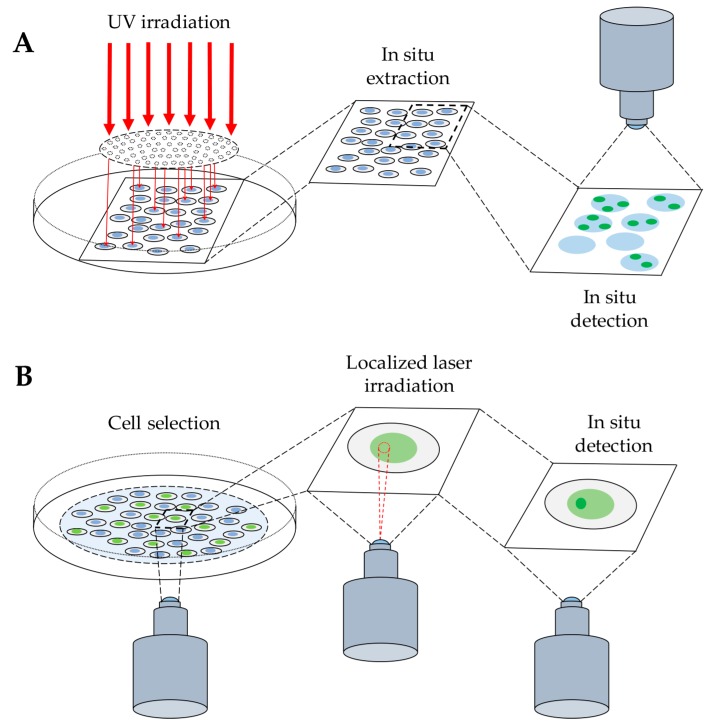
Detection of protein binding to DNA after radiation-induced DNA damage. (**A**) Schematic representation of cell exposure to UV radiation through a micropore polycarbonate filter to produce local DNA damage (see Section 2.1.1). The next step of in situ extraction allows the release of unbound molecules, enabling the final detection of protein bound to DNA, by immunofluorescence technique. (**B**) Cell exposure to local laser irradiation to detect protein binding to DNA by indirect immunofluorescence method, as above, or by direct visualization of the accumulation of a fluorescently-tagged protein at the site of DNA damage in living cells (see Section 2.2).

**Figure 2 ijms-20-05736-f002:**
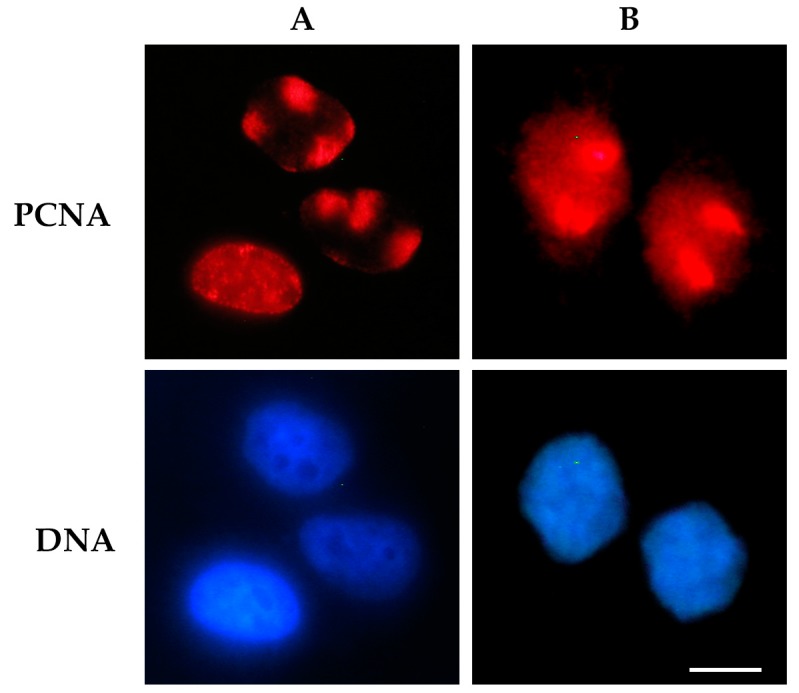
Detection of proliferating cell nuclear antigen (PCNA) protein bound to DNA after UV radiation-induced DNA damage. Cells were exposed to UV radiation (30 J/m^2^) through a 3 μm micropore polycarbonate filter to produce local DNA damage. In (**A**), the cells were processed for an in situ extraction to visualize only PCNA bound to DNA at local damage sites (red spots), or at sites of DNA replication (diffuse pattern) [24]. In (**B**), cells were directly fixed after UV irradiation to detect the total nuclear PCNA. In both panels, PCNA protein (red fluorescence) was detected by immunofluorescence staining with PC10 monoclonal antibody and an Alexa 594-conjugated secondary antibody. DNA was stained with Hoechst 33258 dye. Scale bar = 10 μm.

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
