# Peer review of "In Situ Analysis of DNA-Protein Complex Formation upon Radiation-Induced DNA Damage"

_ijms, 2019, doi:10.3390/ijms20225736_

Round 1
Reviewer 1 Report
In this well-focused review article the authors critically review methods aimed at detecting the formation of protein complex at sites of ionizing and UV radiation-induced damage to DNA with emphasis on cell imaging methods. The latter approaches provide insightful information on spatio-temporal dynamics recruitment of proteins implicated in the DNA damage response (DDR) that includes recognition of the lesions and their repair. Limitations and caveats of the two main classes of methods that consist of immunofluorescence- and probes-based techniques are also discussed in details. The addressed topic that is highly relevant should be of interest for the readership of the Journal. However a few clarifications and minor corrections are required as outlined below.
Main remarks
1 - Insufficient information on radiation-induced DNA damage. The authors mentioned a few types of DNA lesions without providing enough basic information that is necessary for non-experts in the field. It may be lndicated that UVB radiation acts mostly through direct excitation of pyrimidine bases giving rise to bipyrimidine lesions including mostly cyclobutane pyrimidine dimers (CPDs) and to a lesser extent pyrimidine (6-4) pyrimidine dimers (Cadet et al, Cold Spring Harb Perspect Biol 2013, 5, a012559, Photochem Photobiol Sci 2018, 17, 1816-41). UVA radiation has been shown to generate through photon absorption CPDs and oxidatively damaged DNA as the result of photosensitization reaction. Ionizing radiation acts in a less specific way through two main processes including ionization and hydroxyl radical mediated oxidation reactions leading to numerous lesions consisting, in addition to DNA-protein crosslinks, of single, tandem and clustered lesions that involved all bases and/or the deoxyribose moiety (Goodhead, Int J Radiat Biol 1994, 65, 7-17; Cadet et al, Free Radic Biol Med 2017, 107, 13-34). The authors should be more precise when they discussed data concerning UV radiation by mentioning either UVB or UVA.
2 - Molecular effects of laser UV exposure. It is very important to distinguish effects induced by low intensity UV pulses from those of high intensities UV laser radiation. In the latter case it is well documented that for can lead to bi-photonic reactions including ionization of nucleobases (Angelov et al, Biophys J 2005, 88, 2766-78; Cadet et al, Photochem Photobiol 2019, 95, 59-72). This property may be used to induce cross-links at sites of DNA-protein interactions (see remark 5)
3 - Photo-induced side effects in live-cell imaging studies. It is well documented that toxicological effects may be generated by dye photosensitized reactions to DNA through mostly the generation of singlet oxygen. However, the formation of oxidatively generated damage may trigger the recruitment of dedicated DDR proteins that may interfere with proteins induced by the initial physical DNA insult.
4 - Photochemical versus photophysical effects (pg 8, lns 321-3). It is somewhat confusing to oppose photophysical effects to photochemical effects since photophysical events that precede photochemical reactions are both involve in DNA damage induction. It would be rather preferable to mention thermal effects arsing from deactivation of excited molecules with heat dissipation.
5 - Other methods for visualizing DNA-protein interactions. Thus mapping interactions of proteins with nucleosome at the nucleotide level can be achieved by hydroxyl radical footprinting method associated with nanoscale modeling (Syed et al, Proc Natl Acad Sci USA 2010, 107, 9620-5; JL Charles Richard et al, PLos Genet 2016, 12,, e1006221). The UV laser footprinting method has been applied to map UV laser specific biphotonic lesion that consist of 8-oxo-7,8-dihydroguanine and the monophotonic CPDs in the NF-kB cognate sequence (IN Lone et al, PLos Genet 2013, 12, e10006221).
Minor issues.
Pg 1, ln 34: "pyrimidine-pyrimidone 6-4 photoproducts (6-4Ps)" should be "pyrimidine (6-4) pyrimidone photoproducts (6-4 PPs)".
Pg 7, ln 293: "oxidative lesions" should be "oxidatively generated lesions" since oxidized bases/2-deoxyribose residues do not exhibit oxidizing features (Cooke et al, Chem Res Toxicol 2010, 23, 705-7).
Pg 10, reference 29: "DNA Repair" should be "DNA Repair (Amst)"; the remark applies also to references 41 and 64.
Author Response
We thank this Reviewer for the comments and constructive remarks.
We have taken into considerations all the indicated points and modified accordingly the text (highlighted by red color), as explained in point-by-point answer, and cited relevant literature.
1 - Insufficient information on radiation-induced DNA damage.
We agree that this information was perhaps too summarized and have now added, as requested, basic information on UV radiation (composition and effects produced), (p.1 lines 33-41).
2 - Molecular effects of laser UV exposure.
We have detailed the different effects of low vs high intensity of UV laser exposure (p.8, lines 317-318).
3 - Photo-induced side effects in live-cell imaging studies.
Although some information was already provided, we have indicated the possible consequences of photosensitization, such as the recruitment of other DDR proteins that may interfere with those considered specifically induced by the radiation (p. 6 lines 248-251 and point below).
4 - Photochemical vs photophysical effects.
To comply with this request of clarification, we have rewritten this part mentioning more explicitly the thermal effects as one of the causes inducing cytotoxicity (p. 8 lines 327-331).
5 – Other methods for visualizing DNA-protein interactions.
We acknowledge that there are other relevant methods to detect the above interactions, as those cited by the Reviewer in his comment. However, it must be pointed out that our review has been limited to in situ procedures, as explained in the Introduction (p.2, lines 55-69), thus biochemical approaches, such as the UV laser footprinting method, have been not considered here because outside the purpose of the review.
Minor issues have been corrected as requested.
Reviewer 2 Report
The review describes the in situ procedures for the detection of DNA-protein complex formation following exposure to mutagens, including UV light and ionising radiation. Given that exposure to a variety of mutagens results in a formation of DNA-protein complexes, which represent quite a challenge for the majority of DNA repair pathways, thus substantially contributing to the spontaneous and induced mutagenesis, this review paper addresses the important issue of mutagenesis. The review is well written and comprehensively summarises the results of the key publications. To the best of my knowledge, there is just handful a few reviews addressing this important issue. That is why I am happy to recommend it for publication after minor correction (see my previously submitted comments to the authors).
Page 1, line 29 'improper transmission of genetic information' is highly misleading. It should be changed to mutation in DNA.
Author Response
We thank this Reviewer for its appreciation and the comment indicated.
In particular, the misleading sentence in the Introduction, has been modified according to the suggestion (sentence in red color).